# Interleukin Gene Variability and Periodontal Bacteria in Patients with Generalized Aggressive Form of Periodontitis

**DOI:** 10.3390/ijms21134728

**Published:** 2020-07-02

**Authors:** Petra Borilova Linhartova, Zdenek Danek, Tereza Deissova, Filip Hromcik, Bretislav Lipovy, David Szaraz, Julius Janos, Antonin Fassmann, Jirina Bartova, Ivo Drizhal, Lydie Izakovicova Holla

**Affiliations:** 1Clinic of Stomatology, Faculty of Medicine, Masaryk University, Pekarska 664/53, 60200 Brno, Czech Republic; peta.linhartova@gmail.com (P.B.L.); 375826@mail.muni.cz (F.H.); julo.janos@gmail.com (J.J.); holla@med.muni.cz (L.I.H.); 2Department of Pathophysiology, Faculty of Medicine, Masaryk University, Kamenice 5, 62500 Brno, Czech Republic; 436741@mail.muni.cz (T.D.); lipovy.bretislav@fnbrno.cz (B.L.); 3Clinic of Maxillofacial Surgery, University Hospital Brno, Jihlavska 20, 62500 Brno, Czech Republic; szaraz.david@fnbrno.cz; 4Clinic of Maxillofacial Surgery, Faculty of Medicine, Masaryk University, Jihlavska 20, 62500 Brno, Czech Republic; 5Clinic of Stomatology, St. Anne’s University Hospital, Pekarska 664/53, 60200 Brno, Czech Republic; antonin.fassmann@fnusa.cz; 6Department of Burns and Plastic Surgery, Faculty of Medicine, Masaryk University, Jihlavska 20, 62500 Brno, Czech Republic; 7Department of Burns and Plastic Surgery, University Hospital Brno, Jihlavska 20, 62500 Brno, Czech Republic; 8Institute of Clinical and Experimental Dental Medicine, First Faculty of Medicine, Charles University, Karlovo nam. 554/32, 12808 Prague, Czech Republic; jirina.bartova@post.cz; 9Institute of Clinical and Experimental Dental Medicine, General University Hospital, Karlovo nam. 554/32, 12808 Prague, Czech Republic; 10Department of Dentistry, Faculty of Medicine in Hradec Kralove, Charles University, Simkova 870, 50003 Hradec Kralove, Czech Republic; drizhali@motorenova.cz

**Keywords:** aggressive periodontitis, oral bacteria, inflammation, interleukin, polymorphism, genetic predisposition

## Abstract

Host genetic predispositions to dysregulated immune response can influence the development of the aggressive form of periodontitis (AgP) through susceptibility to oral dysbiosis and subsequent host-microbe interaction. This case-control study aimed to perform a multilocus analysis of functional variants in selected interleukin (*IL*) genes in patients with the generalized form of AgP in a homogenous population. Twelve polymorphisms in *IL-1* gene cluster, *IL-6* and its receptor, *IL-10*, *IL-17A*, and *IL-18* were determined in 91 AgP patients and 210 controls. Analysis of seven selected periodontal bacteria in subgingival sulci/pockets was performed with a commercial DNA-microarray kit in a subgroup of 76 individuals. The pilot in vitro study included stimulation of peripheral blood monocytes (PBMC) from 20 individuals with periodontal bacteria and measurement of IL-10 levels using the Luminex method. Only the unctional polymorphism *IL-10* −1087 A/G (rs1800896) and specific *IL-10* haplotypes were associated with the development of the disease (*p* < 0.05, *P*_corr_ > 0.05). Four bacterial species occurred more frequently in AgP than in controls (*p* < 0.01, *P*_corr_ < 0.05). Elevated IL-10 levels were found in AgP patients, carriers of *IL-10* −1087GG genotype, and PBMCs stimulated by periodontal bacteria (*p* < 0.05, *P*_corr_ > 0.05). We therefore conclude that a combination of genetic predisposition to the altered expression of *IL-10* and the presence of specific periodontal bacteria may contribute to Th1/Th2 balance disruption and AgP development.

## 1. Introduction

Aggressive phenotypes of periodontitis (grade 3 periodontitis) are marked with fast progression (more than 2 mm of bone resorption over 5 years), destruction not commensurate with age and biofilm deposits, or typical extent and specific clinical pattern [1]. Historically, the terms early-onset or juvenile periodontitis (for patients with disease onset under 35 years of age) were used; lately, the term aggressive periodontitis (AgP) is preferred [2,3]. The underlying mechanisms of periodontitis are now believed to be the same for both the slow (formerly called chronic periodontitis, CP) and rapid progression forms [3].

Periodontitis is a multifactorial biofilm-induced host-mediated oral disease characterized by the chronic inflammation of gums, progressive bone loss and, eventually, loss of teeth [4]. Potential keystone periodontal bacteria, especially *Aggregatibacter actinomycetemcomitans* and *Porphyromonas gingivalis*, could promote the disruption of tissue homeostasis and the subsequent dysbiotic change in the subgingival biofilm [5]. Specific pathogen-associated molecular patterns (PAMP), such as lipopolysaccharides (LPS) of these Gram-negative bacteria [6,7], and bacterial virulence factors stimulate an inflammatory host response via regulation of cytokine expression [8].

Cytokines, including interleukins (IL), play an important role as cell-signalling molecules in the regulation of inflammation, which should eliminate bacteria with pathogenic potential and stimulate the regeneration of affected tissues. However, dysfunction of immunoregulation leads to a prolonged inflammatory response, formation of granulation tissue, and extensive resorption of alveolar bone [9,10]. The inflammatory state of the tissue itself may induce an ecological change facilitating the shift in the oral microbiota composition and potentially driving the excessive growth of periodontal bacteria [11]. The exact mechanisms that cause such dysfunction are not fully understood. Review of the literature did not show any qualitative differences in the oral microbiota and immunological aspects between AgP and CP; however, the neutrophil function may be more genetically conditioned in AgP than CP [12]. Moreover, individual behavioral factors, such as smoking or oral hygiene levels, largely affect the ability of periodontal tissues to re-establish the host-biofilm equilibrium [13].

Family-based studies have shown a relatively high risk for relatives of AgP patients to also suffer from this disease, the percentage of affected siblings may reach 50% [14,15]. The familial aggregation suggests a significant genetic component in the increased predisposition to this disease. The hereditary pattern for AgP is autosomal-dominant with reduced penetrance [16]. Evidence supports the contribution of a few major genes or multiple small-effects genes [17]; however, the gene–gene and gene–environment interactions are also important [18]. While novel genetic risk factors for AgP have been identified in genome-wide association studies [19], most genetic research into periodontitis has focused on the known gene polymorphisms that play a role in the regulation of the immune system, tissue destructive processes, or metabolism mechanisms.

The candidate genes for AgP include, among others, *IL-1* and its receptor antagonist (*IL-1RN*), *IL-6* and its receptor (*IL-6R*), *IL-10*, *IL-17A*, and *IL-18*. As individual case-control studies in various populations reported controversial results, meta-analyses focused on variants in the mentioned genes were performed and their relation to localized, generalized, or both AgP was predicted [20,21,22,23,24,25,26,27,28,29,30]. Studies investigating combinations of polymorphisms within specific haplotypes and host genotypes with microbial profiles might provide important information on the pathogenesis of this complex disease [31].

It is necessary to point out that most published studies focus on single variants only and have been performed (most likely due to a low prevalence of this disease) on small non-homogeneous populations combining various forms of AgP (localized vs. generalized; see Appendix A). Meta-analyses therefore also include single variants only and combine both above-mentioned types of AgP (in some instances even including chronic periodontitis). Besides, the genetic diversity of the populations, especially in meta-analyses, makes the interpretation of such results more difficult. For this reason, we considered it necessary to perform a genetic association study simultaneously evaluating multiple interleukin-coding genes in a larger genetically homogenous population, such as is the population of the Czech Republic (a vast majority of the population is Caucasian). However, so far gene variants in only one *IL* (*IL-8*) were studied in the AgP patients studied on this population [32].

The aims of the current study were: (i) to provide a multilocus and haplotype analysis of functional variants in the selected genes IL-encoding genes ILs in patients with the generalized form of AgP and to compare it to healthy controls in a genetically homogenous population. We hypothesized that a multilocus and haplotype analysis in a large sample of homogenous population could provide a more complex view of the susceptibility to the aggressive form of periodontitis; (ii) to compare the presence and composition of periodontal bacteria between subgingival sulci/pockets in generalized AgP patients with generalized AgP and in healthy controls; (iii) to build on these genetical and microbial findings, to perform and design an additional pilot in vitro study, in which monocytes from individuals with known immunogenetic profiles would be stimulated by periodontal bacteria and specific IL production would be evaluated; (iv) to propose a model explaining the association between the genetic variants in *IL-10* and development of AgP based on the extensive literature review and on the results of the experimental parts of this study.

## 2. Results

301 unrelated adult subjects from the Czech Republic were enrolled in this case-control study: 91 patients (mean age ± standard deviation, SD: 26.8 ± 7.3 years) with periodontitis and 210 healthy controls (33.3 ± 12.4 years) (*p* < 0.001). There were no significant differences between the cases and the controls relating to the male/female ratio (39.6%/60.4% vs. 47.1%/52.9%, respectively; *p* > 0.05) or smoking status (32.8% vs. 25.0% smokers, respectively; *p* > 0.05).

### 2.1. Genetic Analysis

All studied polymorphisms were in the Hardy–Weinberg equilibrium (HWE) in the control group (*p* > 0.05). There were no significant differences in the allele and genotype distributions of any of the studied single-nucleotide polymorphisms (SNP) between patients with AgP and controls (*p* > 0.05).

Nevertheless, the *IL-10* −1087GG (rs1800896) genotype (vs. AG + AA genotypes) was more frequent in patients with AgP than in controls (*p* = 0.02, *P*_corr_ > 0.05), see Table 1. However, no significant associations among clinical parameters (e.g., probing depth—PD) and gene polymorphisms were found.

The haplotype analysis was performed for *IL-1*, *IL-10* and *IL-18* gene polymorphisms. Frequencies of all haplotypes between cases and controls (as calculated by a multiple model) were similar, see Table 2. Nevertheless, in the recessive model, the *IL-10* GCC (rs1800896/rs1800871/rs1800872) haplotype was associated with AgP (*p* = 0.02, OR = 2.07, 95% CI = 1.13–3.78, *P*_corr_ > 0.05) and in the dominant model, the *IL-18* AGC (rs1946518/rs187238/rs4988359) haplotype was present significantly more often in patients with AgP (*p* = 0.046, OR = 1.84, 95% CI = 1.02–3.34, *P*_corr_ > 0.05).

In addition, a strong linkage disequilibrium (LD) was observed among *IL-10* polymorphisms (rs1800896/rs1800871/rs1800872) in controls, see Figure 1.

### 2.2. Determination of the Selected Oral Bacteria

In the subgingival plaque of patients with AgP, *Tanarella forsythia*, *P. gingivalis*, *Treponema denticola*, and *Parvimonas micra* occurred more often in concentrations higher than 10^3^ colony-forming units (CFU) than in controls (*p* < 0.001, *P*_corr_ < 0.05). No association of the presence of *A. actinomycetemcomitans*, *Prevotella intermedia*, and *Fusobacterium nucleatum* with AgP was found (*p* > 0.05), see Table 3. No significant difference in the occurrence of individual bacteria was found between smokers and non-smokers either in the control group or in the AgP group (*p* > 0.05). An analysis of non-smokers only revealed a higher occurrence of *T. forsythia*, *P. gingivalis* and *T. denticola* patients with AgP than in controls (*p* < 0.01, *P*_corr_ < 0.05). No significant differences were found for *A. actinomycetemcomitans*, *P. intermedia*, and *P. micra* between controls and AgP (*p* > 0.05). Only the subgroup of smokers was not analyzed in detail due to their small representation in the individual subgroups, preventing reliable statistical analysis.

The risk rate for periodontal diseases calculated from the microbial assessment of the presence/absence of all seven investigated bacteria was higher in the patients with AgP than in controls (96.0% vs. 66.7% of patients with increased, high, or very high risk; very high risk was found in 68.0% AgP vs. 13.7% in controls, respectively; *p* = 0.0032).

### 2.3. A Pilot In Vitro Study: IL-10 Levels in Unstimulated/Stimulated PBMCs

Patients with AgP had significantly higher concentrations of IL-10 in unstimulated PBMCs than controls (median (interquartile range, IQR; minimum-maximum without outliers), 6.7 pg/mL (5.6–10.5 pg/mL; 5.0–13.9 pg/mL) vs. 1.1 pg/mL (0.3–3.1 pg/mL; 0.0–6.2 pg/mL), respectively, (*p* = 0.014, calculated by Kruskal–Wallis test)), see Figure 2.

Higher concentrations of IL-10 were found in the PBMCs in four carriers of the *IL-10* −1087GG genotype (rs1800896) than in 13 carriers of *IL-10* −1087AG genotype and three carriers of *IL-10* −1087AA genotype −6.7 pg/mL (5.6–10.5 pg/mL; 5.0–13.9 pg/mL) vs. 1.0 pg/mL (0.2–3.3 pg/mL; 0–6.2 pg/mL) and 1.4 pg/mL (0.5–2.1 pg/mL; 0.5–2.1 pg/mL), respectively (*p* = 0.038, calculated by Kruskal–Wallis test), see Figure 3.

PBMCs stimulated by *A. actinomycetemcomitans* or *T. forsythia* produced more IL-10 than unstimulated PBMCs (*p* < 0.001), while no significant differences between IL-10 levels in PBMCs after stimulation by *P. gingivalis* or *P. intermedia* were found (*P*_corr_ > 0.05), see Table 4. Significant positive correlations in the IL-10 production were found between PBMCs stimulated by pokeweed mitogen (PWM, positive control) and by *T. forsythia*, *P. gingivalis*, or *P. intermedia* (*r*, correlation coefficient, *r* = 0.80, *r* = 0.65, *r* = 0.70, respectively, *p* < 0.01).

## 3. Discussion

Periodontitis is a complex multifactorial disease involving various biological pathways. Periodontal disease has previously been associated with systemic diseases, such as diabetes mellitus, cardiovascular diseases, or rheumatoid arthritis [33,34]. The concept of “leaky mouth”, similarly as leaky gut and leaky brain, suggests that an increase in the permeability of the oral mucosa may allow periodontal bacteria, their toxins, and small molecules produced during inflammation to “leak” into the bloodstream. On the other hand, individuals with altered neutrophil function or counts are unable to mount a successful immune defense barrier against microbial plaque infections and are more susceptible to aggressive forms of periodontal disease [35].

For decades, the periodontal bacterium *A. actinomycetemcomitans* was considered the most likely etiologic agent in AgP; however, now it is regarded as a minor component of the resident oral microbiota and as an opportunistic pathogen in some individuals. In line with our results, bacteria of the red complex (*T. forsythia*, *P. gingivalis*, and *T. denticola*) and the orange complex (*P. micra*) were previously associated with generalized AgP [36,37]. In the European Caucasian population, the total red complex bacteria count in the subgingival plaque was found to be higher in AgP patients than in CP patients, mainly due to *T. denticola*, regardless of the smoking status [38]. It should be noted that there are geographic and ethnic differences in the carriage of periodontitis-associated microorganisms, which need to be taken into account when comparing study reports on periodontal microbiology in different populations [36].

Complex interactions between the microbiota and host genome are at the basis of the susceptibility to periodontitis. According to Clark et al., a hidden factor may lie between the response by the patient’s immune system and the bacterial threat [39]. Recent advances have resulted in a new paradigm shift suggesting that genetically-driven dysbiotic changes in the subgingival microbiota may predispose the patient to a cascade of events leading to the rapid periodontal tissue destruction seen in AgP [5]. Gene polymorphisms may cause phenotypic differences in inflammatory responses, which is important in the individual’s sensitivity to disease, disease progression, or the response to treatment. The systematic review consisting of 13 meta-analyses (71,531 participants) with 25 polymorphisms in seven *IL*s (*IL-1A*, *IL-1B*, *IL-4*, *IL-6*, *IL-8*, *IL-10*, and *IL-18*), *Fcγ* receptors, and other genes for inflammatory mediators, demonstrated that various polymorphisms in *IL*s can be associated with CP and AgP [30].

### 3.1. Interleukin-1 Family

IL-1*α*, IL-1*β*, IL-1RA, and IL-18 belong to the IL-1 family and participate in both innate and acquired immunity. IL-1α, IL-1β, and IL-18 are pro-inflammatory cytokines with several functions contributing to the microbial immune response cascade, while the endogenous IL-1RA inhibitor prevents an acute and chronic overproduction of pro-inflammatory cytokines [40,41].

Polymorphisms in the *IL-1* cluster genes are located on human chromosome 2q12 [42] and are composed of three ligands (*IL-1A*, *IL-1B*, and *IL-1RN*). The *IL-1A* −889TT (rs1800587) genotype creates the consensus site for a transcription factor (Skn-1) in the promotor and significantly enhances the transcriptional activity of the *IL-1A* in comparison to the *IL-1A* − 889CC genotype [43]. Although the *IL-1B* +3953 C/T (rs1143634), which is located in exon five, is a synonymous mutation, the *IL-1B* +3953T allele was associated with a significantly increased production of IL-1β in vitro [44]. The functional relevance of *IL-1RN* intron 2, a variable tandem repeats of 86 bp (VNTR, rs2234663) consists of the number of repeats, which may contain three potential protein binding sites; an α-interferon silencer A, β-interferon silencer B, and an acute phase response element [45]. Although some individual case-control studies associated *IL-1* gene cluster variants with AgP [46,47,48,49], meta-analyses were in line with our findings, that there was no association between *IL-1A* −889 C/T (rs1800587), *IL-1B* +3953 C/T (rs1143634), or *IL-1RN* (VNTR, rs2234663), and susceptibility to AgP, regardless of ethnicity [21,22,23]. In our previous study, we reported that the MAFs of all three polymorphisms in the *IL-1* gene cluster are similar in the Czech CP and AgP patients and in the European Caucasian populations (25.4% vs. 31.9% vs. 28.7% for *IL-1A* −889T allele; 21.4% vs. 25.8% vs. 24.8% for *IL-1B* +3953T allele; and 28.5% vs. 25.8% vs. 21–31% for *IL-1RN**S allele, respectively) [50,51,52,53,54,55,56].

The gene for IL-18 is located on human chromosome 11q22 and contains many polymorphic loci, especially in the promoter region [57,58]. Several different polymorphisms in the promoter region *IL-18* −607 A/C (rs1946518), *IL-18* −137 C/G (rs187238), and in intron 1 *IL-18* −140 C/G (previously −133, rs360721) have been identified [59]. A change at position *IL-18* −607 disrupts a potential binding site for cyclic adenosine monophosphate (cAMP)-responsive element binding protein, while *IL-18* −137C allele has been shown experimentally to disrupt a confirmed H4TF1-binding site [60]. The *IL-18* −607 CC and *IL-18* −137GG genotypes have been associated with enhanced *IL-18* mRNA levels [60]. Evidence suggests that these SNPs and their haplotypes influence the *IL-18* gene expression [61]. The polymorphism *IL-18* −140 C/G (rs360721) is situated in a nuclear factor 1 (NF-1) binding site [59].

Although significant associations between IL-18 gene polymorphisms and an increased risk of AgP were reported in the meta-analysis [29], that analysis included only one case-control study focusing on AgP in which no statistical differences in the genotype and allele frequencies of both *IL-18* −607 A/C (rs1946518) and *IL-18* −137 C/G (rs187238) variants between groups in the German population were recorded [62]. Similar findings were also reported in other studies in German and Italian populations [63,64], only Martelli et al. identified AA/CG (rs1946518/rs187238) haplogenotype as a risk factor for AgP development in the Italian population [65].

Our study did not find any difference between MAFs of SNPs *IL-18* −607 A/C (rs1946518) and *IL-18* −137 C/G (rs187238) in the Czech and European Caucasian populations (39.8% vs. 42.4% for *IL-18* −607A allele, and 31.7% vs. 27.8% for *IL-18* −137C allele, respectively), but the *IL-18* −140G allele was found to be more frequent in the Czech population than in the European Caucasian population (36.4% vs. 21.9%) [66,67,68]. To date, the SNP *IL-18* −140 C/G (rs360721) has not been studied in the context of periodontitis yet.

### 3.2. Interleukin-6

Whereas soluble IL-6R (sIL-6R) activates pro-inflammatory trans-signalling of IL-6, its anti-inflammatory signalling is mediated through the membrane-bound IL-6R. The *IL-6R* polymorphism +48892 A/C (rs2228145, Asp358Ala) located on the human chromosome 1q21 near the disintegrin and metalloproteinase (ADAM) cleavage site enhances the proteolytic cleavage of membrane-bound IL-6R, and elevates sIL-6R levels in blood of carriers of *IL-6R* +48892CC genotype (Ala/Ala) [69,70,71]. The *IL-6* −174 C/G (rs1800795) SNP located on the human chromosome 7p21 in the proximal promoter of IL-6 affects the binding affinity of several transcription factors. The *IL-6* −174G allele, which has higher binding specificity to GATA1 and GATA2 factors, increases the expression of the *IL-6* gene in the cells after stimulation by LPS or IL-1 [72,73,74,75].

Previously, Nibali et al. determined *IL-6* haplotypes in twelve non-smoking UK Caucasian patients with AgP and associated them with responsiveness to clinical presentation, magnitude, and kinetics of local and systemic inflammatory responses following non-surgical and surgical periodontal therapy [76]. In a meta-analysis including four case-control studies, the *IL-6* −174G allele was found to be a risk factor for AgP, but not for CP development [24]. Although the findings regarding *IL-6* −174 C/G (rs1800795) and AgP in the Turkish population are controversial [77,78], no association of this SNP with AgP was found in the Italian population, which is in accordance with results of our previous study on the Czech population [64]. Interestingly, haplotypes formed by several *IL-6* SNPs including *IL-6* −174 C/G (rs1800795) displayed significant association with the localized AgP but not with the generalized AgP phenotype in Caucasians [79]. Development of CP has not been previously associated with *IL-6* −174 C/G (rs1800795) in the Czech population [80]. No differences in MAFs of both *IL-6* −174 C/G (rs1800795) and *IL-6R* +48992 A/C (rs2228145) were observed between the Czech and European Caucasian populations (43.1% vs. 41.5% for *IL-6* −174C allele, and 39.8% vs. 36.0% for *IL-6R* +48992C allele, respectively) [81,82]. To our knowledge, *IL-6R* +48992 A/C (rs2228145) have not been previously studied in patients with periodontal disease.

### 3.3. Interleukin-17

IL-17, a cytokine produced by Th17 cells, has a pro-inflammatory effect as it stimulates production of inflammatory mediators, such as IL-1, IL-6, and tumor necrosis factor-alpha (TNF-α), in various cell types [83]. The promotor SNP *IL-17A* −197 A/G (rs2275913) is located on the human chromosome 6p12. The *IL-17A* −197A allele displayed a higher affinity for the nuclear factor of activated T cells (NFAT), a critical transcription factor involved in IL-17 regulation. In vitro stimulated T cells from healthy individuals possessing the *IL-17A* −197A allele produced significantly more IL-17 than those without this allele [84].

Controversial findings were reported in Indian and Brazilian populations, while Chaudhari et al. associated *IL-17A* −197A allele with localized AgP [85] and Saraiva et al. found G allele to be a risk factor for the generalized form of this disease [86]. Their studied populations were almost twice to three times smaller than that in our study and there is also a difference in the MAFs among populations (68.6% in Indian AgP patients, 20.0% in Brazilian AgP patients). A meta-analysis of these two studies reported no association between *IL-17A* −197 A/G (rs2275913) polymorphism and AgP [28], similarly as in our study. In comparison to the allele and genotype frequencies, there are no differences in distribution between 244 CP patients and AgP patients (13.1% vs. 12.1% for *IL-17A* −197AA genotype, MAF 33.0% vs. 30.6% for *IL-17A* −197A allele, respectively) [87]. In line with our results, the MAF in European Caucasian population is 38.0% [88].

### 3.4. Interleukin-10

IL-10, a carrier of strong anti-inflammatory properties, plays a crucial role in limiting the immune response to pathogens; thus, it prevents damage to the host and maintains the tissue homeostasis. Macrophages are the most important producer of IL-10 although T helper 1 (Th1) and Th2 lymphocytes, dendritic cells, cytotoxic T cells, B lymphocytes, monocytes, and mast cells can also produce it to some extent. IL-10 inhibits the expression of the major histocompatibility complex (MHC) class II as well as the co-stimulatory molecules; in this way, it downregulates the expression of *IL-1*, *IL-6*, *IL-8*, *IL-12*, and *TNF-α* [89]. Dysregulation of IL-10 is associated with enhanced immunopathology in response to infection as well as with the increased risk for development of many autoimmune diseases [90].

The *IL-10* gene is located on the human chromosome 1q31–32 [91]. Three SNPs in the promoter region, the *IL-10* −1087 A/G (rs1800896), *IL-10* −824 C/T (rs1800871), and *IL-10* −597 C/A (rs1800872), of the *IL-10* gene have been shown to alter *IL-10* mRNA and protein levels [92,93]. These variants, which form three predominant haplotypes (GCC, ACC, ATA) are in strong LD. The ATA/ATA (rs1800896/rs1800871/rs1800872) haplogenotype was associated with the lowest expression of *IL-10* [94]. Transcription factors PU.1 and Spi-B bound to both *IL-10* −1087G and −1087A alleles, whereas the transcription factor Sp1 only bound to the −1087G allele, which might contribute to higher levels of IL-10 production in B cells [93]. Enhanced expression of *IL-10* may be also mediated by greater affinity of transcription factor poly (ADP-ribose) polymerase-1 (PARP-1) for the *IL-10* −597C allele in their carriers [95].

A meta-analysis of 26 studies indicated that the *IL-10* −1087AA genotype in the Caucasian population and the *IL-10* −1087GG genotype in the Han population might be putative risk factors for CP, while no association with AgP development was found [96]. In another meta-analysis, significant associations between IL-10 −597A/C (rs1800872) polymorphism and the risk of AgP in the non-Asian populations were observed in particular genetic models, while the allele and genotype frequencies of two other IL-10 SNPs were found similar between AgP patients and controls [26].

Our results of the *IL-10* gene variability and AgP susceptibility are controversial, because the *IL-10* −1087GG genotype vs. AA+AG genotypes and also *IL-10* GCC haplotype were found to be more common in AgP patients than in controls. In the context of *IL-10* −1087 A/G (rs1800896), three case-control studies in Europe, namely, on UK (51 AgP patients vs. 100 controls), German (32 AgP patients vs. 34 controls), and Italian populations (122 AgP patients vs. 246 controls) were performed. While Brett et al. and Scapoli et al. found similar allele and genotype frequencies between cases and controls; *IL-10* ATA/ATA haplogenotype was associated with generalized AgP in the German population [97,98,99,100]. This finding is disputable given the very small number of individuals involved in the haplogenotype analysis. On the other hand, the haplotype approach seems to be more promising than studying single gene variants. The MAFs of all three polymorphisms in *IL-10* do not differ much between AgP patients in the Czech and European Caucasian populations (44.8% vs. 45.3% for *IL-10* −1087G allele, 24.3% vs. 24.0% for *IL-10* −824T allele, and 23.3% vs. 32.4% for *IL-10* −597A allele, respectively) [101,102,103].

### 3.5. A Pilot In Vitro Study

In the inflammatory process, the aggregation of inflammatory cells (fibroblasts, endothelial cells, tissue macrophages, mast cells, monocytes, lymphocytes, and neutrophils) at the site of the tissue affected by an infection is initiated by a number of soluble mediators, such as cytokines. Cytokines are pleiotropic endogenous inflammatory and immunomodulating signalling molecules. Upon activation by an extensive range of agents, e.g., bacterial cell wall products (LPS), endotoxins, and cytokines, nuclear factor kappa B (NFκB) acts to induce gene expression of many cytokines involved predominantly in inflammation and other processes. In the early stages of inflammation, TNF-α and IL-1 are the first cytokines to be released and promote the secretion of IL-6. After some time, compensation mechanisms arise to dampen the pro-inflammatory response, such as IL-10, IL-1RA. Active IL-10 is secreted by CD4^+^ Th2 cells, Treg, monocytes, and macrophage cells of the immune system. IL-10 controls inflammatory processes by suppressing the expression of pro-inflammatory cytokines such as TNF-α, IL-1, IL-6, chemokines, adhesion molecules, as well as antigen-presenting and co-stimulatory molecules in monocytes/macrophages, neutrophils, and T cells [104,105,106,107].

The findings of the pilot in vitro study supported the results of our genetic association study in the Czech population: higher levels of IL-10 produced by PBMCs were found in AgP patients than in controls; in carriers of the *IL-10* −1087GG genotype, enhanced levels were also detected when compared to the carriers of other genotypes. Similarly, increased IL-10 production was found in PBMCs stimulated by *A. actinomycetemcomitans* or *T. forsythia* than in unstimulated PBMCs. Our findings are in line with the functional significance of this SNP as well as with a Danish study where slightly higher IL-10 levels were found in plasma and unstimulated whole blood cell cultures of AgP patients with the generalized form than in controls [93,108]. Based on our findings and literature review, we suggested a possible model of *IL-10* gene variability involvement in the etiopathogenesis of AgP, see Figure 4.

It is known that both overexpression (e.g., in systemic lupus erythematosus or tuberculosis) and deficiency (e.g., in psoriasis or rheumatoid arthritis) of IL-10 can be associated with pathological processes [89]. It is assumed that abnormal cytokine production can speed up the periodontitis progression and that the control of the Th1/Th2 balance is crucial for the immunoregulation of the periodontal disease [109,110]. Nibali et al. have recently suggested the tendency that “anemia of inflammation” represents, besides leukocytosis, another typical feature of periodontitis [111]. Genetic and microbial markers, as well as those associated with the host response, help in differentiating between individual phenotypes of periodontitis as well as in understanding of the initiation and progression of periodontitis. Their further research is therefore of utmost importance [3].

The strengths of our study are: (1) Strict criteria for inclusion/exclusion of individuals to the study were applied, which allowed a comparison of extreme phenotypes only (cases with the generalized form of AgP vs. healthy controls) without the presence of systemic disease. The study was conducted in a relatively homogenous population. (2) Multiple gene variants with functional consequences were studied, haplotype analyses were performed. This is the first study investigating *IL-6R* +48992 A/C (rs2228145) and *IL-18* −140 C/G (rs4988359) polymorphisms in AgP patients. (3) The results obtained both in the microbial and genetic analysis were further explored and confirmed in the subsequent pilot in vitro study on human PBMCs.

The limitations of this study are: (1) A small sample size; on the other hand, the sample size is comparable with similar genetic association studies conducted in European populations in this context [62,63,64,97,98,99,100]. (2) The mean age of controls was significantly higher than in AgP patients. While the differences in mean age between groups should not affect the results of the genetic association study and the pilot in vitro study, the findings of microbial analysis should be, in view of that fact, considered carefully. (3) The present study is limited by the fact that levels of ILs in circulation or in the gingival tissue were not measured and the presence of the selected periodontal bacteria was examined only in the 76 individuals. This was, however, caused by a necessity to apply stricter exclusion criteria (in particular, the use of ATBs was a limiting factor); besides, we believe that 76 patients constitute a sufficient population for this type of study.

## 4. Materials and Methods

The study was performed with the approval of the Ethics Committees of the Faculty of Medicine, Masaryk University Brno (No. 15/2009, approved on 21 December 2009; No. 13/2013, approved on 20 May 2013) and St. Anne’s Faculty Hospital (No. without number/2005, approved on 30 June 2005). A written informed consent was obtained from all participants in line with the Declaration of Helsinki before inclusion in the study.

### 4.1. Study Design, Clinical Examination and Sample Collection

The case-control genetic association and microbiological studies were conducted in the laboratory of the Department of Pathophysiology, Faculty of Medicine, Masaryk University, Brno, Czech Republic, in the period from 2005 to 2015. The pilot in vitro study focusing on IL-10 levels in stimulated/unstimulated PBMCs was performed in the laboratory of the Institute of Clinical and Experimental Dental Medicine, General University Hospital and First Faculty of Medicine, Charles University, Prague, Czech Republic, in the period from 2013 to 2014.

The periodontal status was evaluated in individuals recruited from the patient pool of the Clinic of Stomatology, St. Anne’s Faculty Hospital Brno (A.F.) and Department of Dentistry, Faculty of Medicine in Hradec Kralove, Charles University (I.D.). The diagnosis of periodontitis/non-periodontitis was based on detailed clinical examination, medical and dental history, tooth mobility, and radiographic assessment as described in our previous study [32] in line with criteria defined by the World Workshop on the Classification of Periodontal Diseases and Conditions [112].

The inclusion criteria for all individuals were: older than 18 years, willingness to participate in the study, Czech nationality, Caucasian race, general good health. The exclusion criteria for all individuals were: genetic relatedness between individuals, other nationality/race than Czech Caucasian, previous periodontal treatment, current pregnancy or lactation, drug abuse, history of systemic diseases such as cardiovascular disorders (e.g., coronary artery diseases), diabetes mellitus, malignant diseases, immunodeficiency, and autoimmunity.

The specific inclusion criteria for cases were: only patients with age at disease onset <35 years, attachment loss of 4 mm or more in at least 30% of the teeth (at least three of the affected teeth were neither first molars nor incisors), and the severity of the attachment loss being inconsistent with the amount of dental plaque. The specific inclusion criteria for controls were: at least 20 remaining teeth, no signs of periodontitis, values of the Community Periodontal Index of Treatment Needs (CPITN) <3 [113]. The specific exclusion criteria for the microbial analysis were: antimicrobial drug therapy or professional dental hygiene two months or less before sampling. The specific exclusion criteria for the pilot in vitro study on PBMCs were: antimicrobial drug therapy less than two months before sampling, and former or current smoking.

The smoking status was recorded; individuals who never smoked were referred to as non-smokers, and both former and current smokers were referred to as smokers.

Subgingival plaque samples from the deepest pocket in periodontitis patients (and from the deepest sulcus in healthy subjects) of each quadrant were collected for microbial analysis before subgingival scaling [32]. Sampling was performed using a paper cone (ISO 40, VDW GmbH, Munich, Germany) inserted into the base of the pocket for 20 s until it was covered with biological material up to 2/3. The paper cone was placed using tweezers into a sterile tube and stored at −20 °C.

In all individuals, 9 mL of peripheral blood samples were collected into a tube with ethylenediaminetetraacetic acid (EDTA) for DNA isolation. Samples of blood were stored at −20 °C. In a subgroup of the randomly selected individuals included in the in vitro study, 20 mL of peripheral blood were collected into a tube with heparin for PBMCs isolation. Samples of blood were stored at −80 °C.

### 4.2. Isolation of Genomic DNA and Genetic Analysis

The genomic DNA for genetic analysis was extracted from the peripheral blood leukocytes using standard phenol/chloroform procedures with proteinase K. Concentration of working samples was 50 ng/μL.

Twelve polymorphisms in the *IL-1* gene cluster, *IL-6* and its receptor, *IL-10*, *IL-17A*, and *IL-18* were studied by methods based on polymerase chain reaction (PCR). Detailed protocols for the determination of *IL-1A* −889 C/T (rs1800587), *IL-1B* +3953 C/T(rs1143634), and *IL-1RN* VNTR [114], *IL-6* −174 C/G (rs1800795) [80], *IL-10* −1087 A/G (rs1800896), *IL-10* −824 C/T (rs1800871), and *IL-10* −597 C/A (rs1800872) [115], and *IL-17A* −197 A/G (rs2275913) [116] were published previously.

Amplification of DNA fragments containing *IL-6R* +48892 A/C (rs2228145) polymorphism was carried out in a reaction volume of 25 μL containing 75 ng of genomic DNA, 0.24 μM of each primer (forward primer: 5′-GTTAAGCTTGTCAAATGGCCTGTT-3′ and reverse primer: 5′-CAGAGGAGCGTTCCGAAG-3′) previously designed by Galicia et al. [117], 1.5 U of DNA polymerase (Thermo Fischer Scientific, Waltham, MA, USA), 3.5 mM of MgCl_2_, 10× MgCl_2_-free reaction buffer with (NH_4_)_2_SO_4_, and 0.2 mM deoxyribonucleoside triphosphate mix (dNTP, Thermo Fischer Scientific, Waltham, MA, USA). Denaturation for 5 min at 95 °C was followed by 25 cycles at 95 °C for 1 min, 57 °C for 1 min, and 72 °C for 1 min in a Sensoquest labcycler (Schoeller Instruments, Prague, Czech Republic). The last synthesis step was extended to 10 min at 72 °C. The PCR products were then digested with *Hinf*I restriction enzyme. The restriction was performed in a volume of 30 μL consisting of 25 μL of the PCR product, 10× CutSmart Buffer (New England Biolabs, Ipswich, MA, USA), and 5 U of *Hinf*I enzyme (New England Biolabs, Ipswich, MA, USA), and incubated overnight at 37 °C. The amplicon and digested products: 188 + 70 bp (AA), 258 + 188 + 70 bp (AC), or 258 bp (CC) fragments were electrophoresed in 2% agarose gel with ethidium bromide staining.

The *IL-18* −607 C/A (rs1946518) and *IL-18* −137 C/G (rs187238) were genotyped using an allele-specific polymerase chain reaction (AS-PCR) according to Giedraitis et al. [60] with modifications of the PCR reaction conditions. *IL-18* −607 C/A (rs1946518) was carried out in a reaction volume of 15 μL containing 100 ng of genomic DNA, 0.6 µM reverse primer and allele specific primer (reverse primer: 5′-TAACCTCATTCAGGACTTCC-3′, and specific forward primers: 5′-GTTGCAGAAAGTGTAAAAATTATTAA-3′ or 5′-GTTGCAGAAAGTGTAAAAATTATTAC-3′), 0.3 µM control primer (control forward primer 5′-CTTTGCTATCATTCCAGGAA-3′), 20 mM Dynex buffer, 6.7 mM MgCl_2_, 4 mM dNTPs, and 1 U *Taq* polymerase (Thermo Fischer Scientific, Waltham, MA, USA). For each specific primer, a separate PCR reaction was prepared. Initial denaturation at 94 °C for 2 min was followed by seven cycles at 94 °C for 20 s, 64 °C for 40 s, 72 °C for 40 s, and 25 cycles at 94 °C for 20 s, 57 °C for 40 s and 72 °C for 40 s. The 301 bp control PCR product and 196 bp specific PCR products were visualized by 2% agarose gel electrophoresis with ethidium bromide staining.

*IL-18* −137 C/G (rs187238) was determined using 100 ng genomic DNA, 0.5 µM reverse primer and allele-specific primer (reverse primer: 5′-AGGAGGGCAAAATGCACTGG-3′, and specific forward primers 5′-CCCCAACTTTTACGGAAGAAAAC-3′or 5′-CCCCAACTTTTACGGAAGAAAAG-3′), 0.3 µM control primer (control forward primer: 5′-CCAATAGGACTGATTATCCGCA-3′), 20 mM Dynex buffer, 10.0 mM MgCl_2_, 4 mM dNTPs, and 1 U *Taq* polymerase in the total 15 µL PCR volume. The steps of the PCR reaction were as follows: denaturation at 94 °C for 2 min, five cycles of 94 °C for 20 s, 58 °C for 60 s, and then 25 cycles of 94 °C for 20 s, 50 °C for 40 s, 72 °C for 40 s, and a final extension at 72 °C for 5 min. The 420 bp control PCR product and 261 bp specific PCR products were visualized by 2% agarose gel electrophoresis stained by ethidium bromide.

As positive controls for the above-mentioned polymorphisms (*IL-18* −607 C/A (rs1946518) and *IL-18* −137 C/G (rs187238)), DNA samples with the known genotypes obtained by Sanger sequencing were used. Forward primer: 5′-GCAGAGGATACGAGTACTTCTTTTA-3′ and reverse primer: 5′-TTGCCCTCTTACCTGAATTTTG-3′ for Sanger sequencing were created by Primer3 program tools. PCR reaction consisted of: 20 mM (NH_4_)_2_SO_4_, 3 mM MgCl_2_, 0.5 µM of each primer, 2 mM dNTPs, and 3 U *Taq* polymerase in 50 µL PCR volume. The first step of PCR was denaturation: at 95 °C for 5 min, followed by 38 cycles at 95 °C for 40 s, 55.5 °C for 45 s and 72 °C for 45 s, with a final extension at 72 °C for 5 min. The 596 bp length PCR product was visualized by 1% agarose gel electrophoresis stained by ethidium bromide. Next were ExoSAP-ITTM purification, sequencing PCR reaction, another purification with 100% ethanol and EDTA, drying up and dissolution in Hi-Di formamide. All procedures were conducted according to the manufacturer’s instruction: “Generating high-quality data using the Big Dye™ Terminator v3.1 Cycle Sequencing Kit” (MAN0015798, Rev. A.0, Applied Biosystems). Sequencing was done on the sequencer ABI 3130-Avant genetic analyzer (Applied Biosystems) with a 36 cm long capillary (Applied Biosystems) filled with NimaPOP7 (Nimagen). Injection at 1.2 kV lasted 12 s. Electrophoresis was performed at 8.5 kV, 60 °C, and 5 μA for 4000 s. The laser was set at a constant power of 15 mW. Data were collected by Data Collection program (version 3.0, Applied Biosystems) and results were analyzed using Sequencing Analysis program (version 5.4, Applied Biosystems).

For genotypization of *IL-18* −140 C/G (rs4988359), restriction fragment length polymorphism PCR (RFLP-PCR) was used. The primer sequence was previously designed by Kruse et al. [59]. PCR was carried out in a volume of 15 μL containing 100 ng of DNA, 0.5 μM of each primer, 3 mM of dNTPs and 1 U of *Taq* DNA polymerase with the 20 mM KCl buffer and 4 mM MgCl_2_ (Thermo Fischer Scientific, Waltham, MA, USA). After denaturation for 2 min at 94 °C, 35 cycles were carried out, with each cycle consisting of 30 s at 94 °C, 30 s at 50 °C and 30 s at 72 °C, and 5 min at 72 °C for the final extension. Restriction was performed in a volume of 30 μL containing 15 μL of the PCR product, 10 U restriction enzyme *Sma*I from NEB (Frankfurt, Germany) and the 10× buffer T and bovine serum albumin (BSA) recommended by the supplier at 30 °C for 12 h. The 243 + 22 bp (GG), 265 + 243 + 22 bp (CG), and 265 bp (CC) fragments were visualized by 3.5% agarose gel electrophoresis with ethidium bromide staining.

### 4.3. Microbial Analysis

Analysis of 7 selected periodontal bacteria (*A. actinomycetemcomitans*, *T. forsythia*, *P. gingivalis*, *T. denticola*, *P. intermedia*, *P. micra*, and *F. nucleatum*) in subgingival sulci/pockets was based on a commercial DNA-microarray kit (Protean Ltd., Ceske Budejovice, Czech Republic). Bacterial colonization was investigated in a subgroup of cases (*N* = 25) and a subgroup of controls (*N* = 51). This test determined the individual bacteria semiquantitatively, the diagnosis of the specific bacterial infection was considered positive when the number of bacterial cells exceeded 10^3^ CFU [32]. The hypothetical risk score of periodontal disease (low/increased/high/very high risk) was calculated by Protean Ltd., Ceske Budejovice, Czech Republic, using their original proprietary algorithm based on the presence and relative quantity of the periodontal bacteria in samples obtained from periodontal pockets/sulci.

### 4.4. PBMCs Isolation, Cultivation and Stimulation by Oral Bacteria In Vitro

PBMCs were isolated from the heparinized blood samples of 20 individuals according to the protocol published previously. PBMCs (10^6^/µL) were stimulated with the selected periodontal bacteria (*A. actinomycetemcomitans*, *T. forsythia*, *P. gingivalis*, and *P. intermedia*) for 3 days [118].

### 4.5. Determination of Interleukin-10 Levels in PBMCs

The medium with PBMCs was collected and stored at −20 °C for the determination of IL-10 levels by the Luminex multiplex method (LUMINEX 100TM analyzer, R&D Systems, USA). The IL-10 levels were measured in PBMCs according to the study by Bartova et al. [118].

### 4.6. Statistical Analysis

Standard methods of descriptive statistics were applied; data are presented as mean and SD for continuous variables and absolute/relative frequencies for categorical variables. The power analysis was performed by Fisher’s exact test assuming standard settings of probability measures, i.e., alpha level < 0.05 and power = 0.80. The allele frequencies were calculated from the observed numbers of genotypes. Differences in allele frequencies were calculated by the Fisher exact test, χ^2^ test was used for comparisons of genotypes and HWE. The value of *p* < 0.05 was considered as statistically significant. The Bonferroni adjustment for multiple testing was applied where appropriate, the adjusted *p* values are denoted as *P*_corr_. Haplotype frequencies were calculated using the SNP Analyzer 2 program (http://snp.istech.info/istech/board/login_form.jsp). Pairwise LD for all possible 2-way comparisons among 3 variants in the *IL-1* gene cluster, 3 polymorphisms in the *IL-10*, and 3 polymorphisms in the *IL-18* in controls was determined according to the Lewontin standardized disequilibrium coefficient D’. One-way analysis of variance (ANOVA) or Kruskal–Wallis ANOVA were performed to compare continuous variables among the groups. Stimulation of PBMCs by periodontal bacteria in comparison with unstimulated samples (negative control) was evaluated by Willcoxon pair test. The correlation between stimulation of PBMCs by PWM (positive control) and periodontal bacteria was tested using Spearman’s coefficient.

Contingency table analysis, ORs, 95% CIs, and significance values were estimated with the use of a statistical program package Statistica v. 13 (StatSoft Inc., Tulsa, Okla., USA).

## 5. Conclusions

For the first time, the *IL-10* −1087GG genotype and *IL-10* GCC haplotype were shown to present risk factors for generalized AgP development. On the other hand, functional polymorphisms in the *IL-1* gene cluster, *IL-6* and its receptor, *IL-17A*, and *IL-18* (and their combinations) were not related to the generalized AgP development in our genetically homogenous population.

Among others, the host genetic predisposition to the overexpression of *IL-10* gene and composition of periodontal bacteria, such as *T. forsythia*, *P. gingivalis*, *T. denticola* and *P. micra*, may trigger the inflammatory response leading to AgP development. A model explaining the relationship between the genetic variants in *IL-10* and development of AgP was suggested.

## Figures and Tables

**Figure 1 ijms-21-04728-f001:**
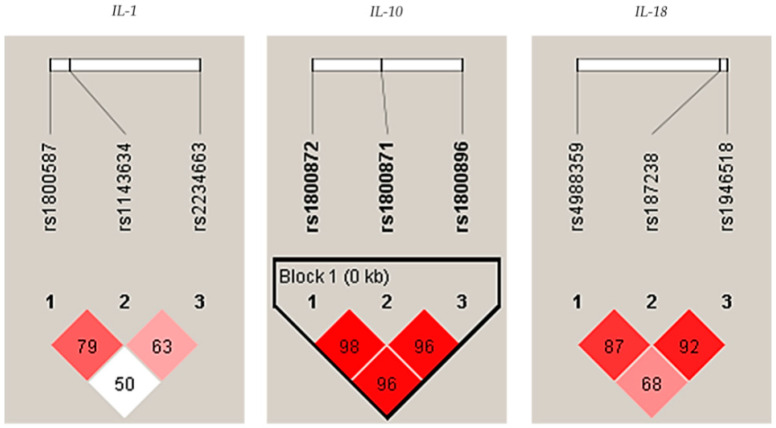
Linkage disequilibrium (LD) plots indicating the D coefficient in the interleukin-1 (*IL-1*), *IL-10*, and *IL-18*, respectively, in controls. Strong LDs are highlighted in red. LD blocks (bold black line), were generated using the solid spine of the LD method.

**Figure 2 ijms-21-04728-f002:**
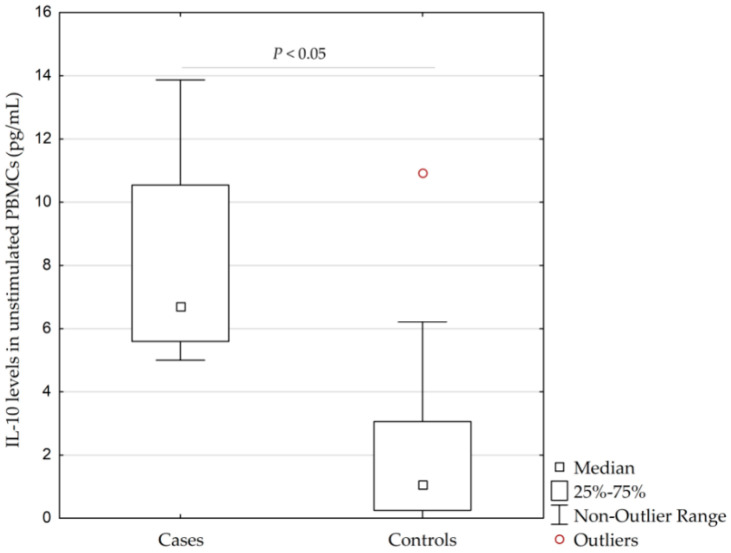
Concentrations of interleukin-10 (IL-10) in unstimulated peripheral blood monocytes (PBMCs) isolated from 4 patients with the aggressive form of periodontitis (AgP) and 16 healthy controls (*p* = 0.014, calculated by Kruskal–Wallis test). IQR, interquartile range.

**Figure 3 ijms-21-04728-f003:**
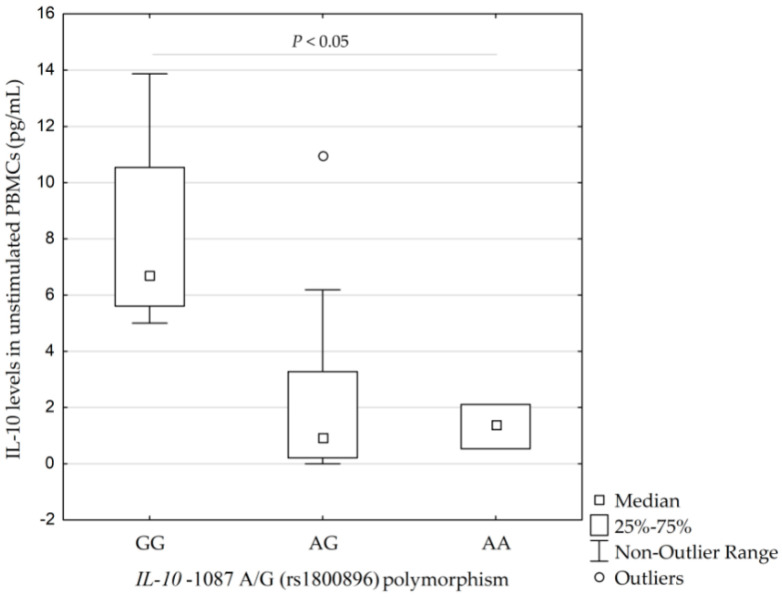
Concentrations of interleukin-10 (IL-10) in unstimulated peripheral blood monocytes (PBMCs) isolated from 20 individuals according to their *IL-10* −1087 A/G (rs1800896) profile (*p* = 0.038, calculated by Kruskal–Wallis test). IQR, interquartile range.

**Figure 4 ijms-21-04728-f004:**
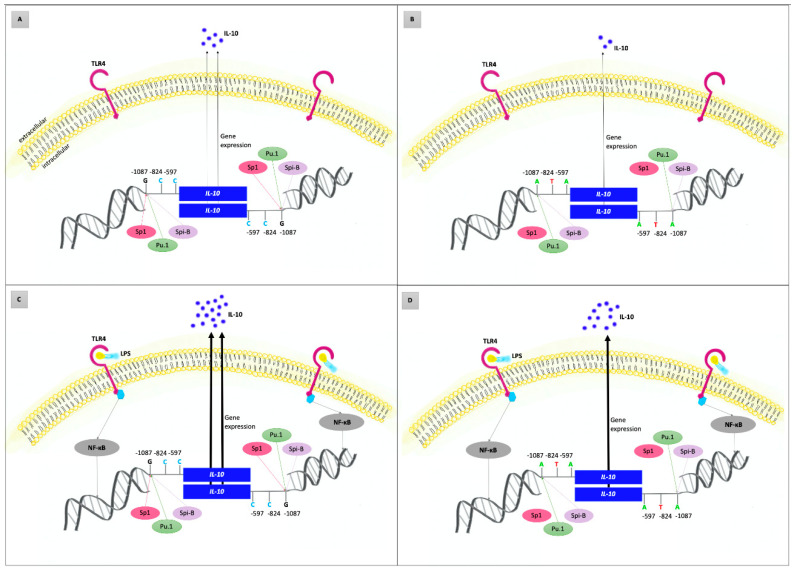
Possible role of interleukin-10 (*IL-10*) gene variants in the etiopathogenesis of aggressive periodontitis (AgP)—model based on the literature review and results of the current study. Concentrations of IL-10 reflect the host genetic predisposition to up/downregulated *IL-10* gene expression and the presence of external stimuli, such as lipopolysaccharides (LPS) released from the outer membrane of Gram-negative bacteria. (**A**,**B**) represent the condition without external stimulation of the immune system cells. (**C**,**D**) show inflammatory status after stimulation of the immune system cells by LPS of periodontal bacteria via toll-like receptor 4 (TLR4) and nuclear factor kappa B (NFκB) leading to the upregulated *IL-10* gene expression and thus massively increased IL-10 production. Genetic predisposition is determined by *IL-10* haplogenotypes, which are composed of three functional promotor gene polymorphisms: *IL-10* −1087 A/G (rs1800896), *IL-10* −824 C/T (rs1800871), and *IL-10* −597 C/A (rs1800872). Carriers of *IL-10* GCC/GCC haplogenotype (**A**,**C**) have −1087G alleles in both strands; therefore, the binding sites for the Sp1 transcription factors are also present in both strands. Under comparable conditions, the *IL-10* gene expression in these individuals is higher than in carriers of the ATA/ATA haplogenotype (**B**,**D**). While the transcription factors PU.1 and Spi-B bind to both −1087G and −1087A alleles, the activator of transcription Sp1 is not bound to *IL-10* −1087A allele. Individuals with the GCC/GCC haplogenotype of the *IL-10* gene and certain periodontal bacteria in periodontal sulci/pockets are more susceptible to the AgP development (Figure 4**C**) than carriers of other *IL-10* haplogenotypes. The disease development may be caused by dysregulation of the cytokine network via *IL-10* overexpression.

**Table 1 ijms-21-04728-t001:** Minor allele frequencies (MAF) and genotype frequencies of selected polymorphisms in interleukins (IL) in patients with the aggressive form of periodontitis (AgP) and healthy controls.

Polymorphism	Genotype MAF	Patients with AgP *N* = 91 (%)	Controls *N* = 210 (%)	*p*	OR 95% CI
*IL-1A* −889 C/T (rs1800587)	CC	46 (50.5)	95 (45.2)		1.00
	CT	32 (35.2)	97 (46.2)		0.68 (0.40–1.16)
	TT	13 (14.3)	18 (8.6)	0.12	1.49 (0.67–3.30)
	T	(31.9)	(31.7)	0.52	1.01 (0.69–1.47)
*IL-1B* +3953 C/T (rs1143634)	CC	51 (56.0)	111 (52.9)		1.00
	CT	33 (36.3)	83 (39.5)		0.87 (0.51–1.46)
	TT	7 (7.7)	16 (7.6)	0.86	1.05 (0.41–2.71)
	T	(25.8)	(27.4)	0.39	0.92 (0.62–1.37)
*IL-1 VNTR*^§^ intron 2, 86 bp VNTR (rs2234663)	LL	51 (56.0)	114 (54.3)		1.00
	LS	33 (36.3)	79 (37.6)		0.93 (0.55–1.58)
	SS	7 (7.7)	17 (8.1)	0.96	0.92 (0.36–2.36)
	S	(25.8)	(26.9)	0.43	0.95 (0.64–1.41)
*IL-6* −174 C/G (rs1800795)	GG	27 (29.6)	69 (32.9)		1.00
	GC	41 (45.1)	101 (48.1)		0.96 (0.54–1.71)
	CC	23 (25.3)	40 (19.0)	0.47	1.47 (0.75–2.90)
	C	(47.8)	(43.1)	0.16	0.83 (0.58–1.17)
*IL-6R* +48992 A/C (rs2228145)	AA	32 (35.2)	76 (36.2)		1.00
	AC	46 (50.5)	101 (48.1)		1.08 (0.63–1.86)
	CC	13 (14.3)	33 (15.7)	0.91	0.94 (0.44–2.01)
	C	(39.6)	(39.8)	0.52	0.99 (0.69–1.42)
*IL-10* −1087 A/G (rs1800896)	AA	23 (25.3)	54 (25.7)		1.00
	AG	44 (48.4)	124 (59.1)		0.83 (0.46–1.51)
	GG	24 (26.4)	32 (15.2)	0.06	1.76 (0.86–3.62)
	AG + AA	67 (73.6)	178 (84.8)	0.02 *	0.50 (0.28–0.91)
	G	(50.5)	(44.8)	0.12	1.26 (0.89–1.79)
*IL-10* −824 C/T (rs1800871)	CC	49 (53.8)	118 (56.2)		1.00
	CT	39 (42.9)	82 (39.0)		1.15 (0.69–1.90)
	TT	3 (3.3)	10 (4.8)	0.74	0.72 (0.19–2.74)
	T	(24.7)	(24.3)	0.49	1.02 (0.68–1.53)
*IL-10* −597 C/A (rs1800872)	CC	50 (54.9)	122 (58.1)		1.00
	CA	38 (41.8)	78 (37.1)		1.19 (0.71–1.98)
	AA	3 (3.3)	10 (4.8)	0.68	0.73 (0.19–2.77)
	A	(24.2)	(23.3)	0.45	1.05 (0.70–1.58)
*IL-17A* −197 A/G (rs2275913)	GG	46 (50.5)	92 (43.8)		1.00
	GA	34 (37.4)	92 (43.8)		0.74 (0.44–1.25)
	AA	11 (12.1)	26 (12.4)	0.53	0.85 (0.38–1.86)
	A	(30.8)	(34.4)	0.23	1.17 (0.81–1.71)
*IL-18* −607 A/C (rs1946518)	CC	25 (27.5)	71 (34.0)		1.00
	AC	54 (59.3)	109 (52.2)		1.41 (0.80–2.46)
	AA	12 (13.2)	29 (13.8)	0.48	0.85 (0.38–1.92)
	A	(42.9)	(39.8)	0.28	1.13 (0.79–1.60)
*IL-18* −137 C/G (rs187238)	GG	45 (49.4)	97 (46.4)		1.00
	GC	42 (46.2)	93 (44.5)		1.03 (0.62–1.71)
	CC	4 (4.4)	19 (9.1)		0.45 (0.15–1.41)
	C	(27.5)	(31.7)	0.20	1.21 (0.82–1.77)
*IL-18* −140 C/G (rs4988359)	CC	39(42.9)	85 (40.7) ^#^		1.00
	GC	45 (49.4)	97 (46.4)		1.01 (0.60–1.70)
	GG	7 (7.7)	25 (12.0)	0.52	0.61 (0.24–1.53)
	G	(32.4)	(36.4)	0.26	1.15 (0.79–1.66)

^#^ Two genotypes missing due to poor quality of isolated DNA; ^§^ L as “long“ allele 412 bp or 326 bp or 498 bp or 584 bp, and S as “short” allele 240 bp; CI, confidence interval; IL, interleukin; *N*, number of individuals; OR, odds ratio. * *p* < 0.05.

**Table 2 ijms-21-04728-t002:** Haplotype frequencies of selected interleukin polymorphisms in patients with the aggressive form of periodontitis (AgP) and healthy controls, and the evaluation of the differences between AgP and control groups (*p* and OR) calculated using the multiple model.

Gene	Haplotype ^‡^			Patients with AgP *N* = 91 (%)	Controls *N* = 210 (%)	*p*	OR 95% CI
*IL-1*	C	C	A	(45.2)	(42.8)	0.54	1.12 (0.79–1.59)
	T	T	A	(20.5)	(21.9)	0.65	0.91 (0.60–1.38)
	C	C	B	(20.0)	(21.6)	0.58	0.89 (0.59–1.35)
	T	C	A	(7.0)	(5.5)	0.58	1.21 (0.62–2.37)
	C	T	A	(1.4)	(2.9)	0.36	0.57 (0.16–2.04)
	T	C	B	(2.0)	(2.7)	0.74	0.77 (0.15–3.84)
	T	T	B	(2.4)	(1.6)	0.31	2.33 (0.47–11.65)
	C	T	B	(1.5)	(1.0)	0.64	1.54 (0.26–9.32)
*IL-10*	G	C	C	(50.5)	(44.4)	0.17	1.27 (0.90–1.81)
	A	C	C	(24.2)	(31.1)	0.09	0.71 (0.48–1.06)
	A	T	A	(23.6)	(22.7)	0.84	0.69 (0.57–1.58)
	A	T	C	(1.1)	(1.2)	0.92	0.92 (0.18–4.80)
	G	T	A	(0.0)	(0.4)	1.00	0.00 (0.00–0.00)
	A	C	A	(0.6)	(0.2)	0.56	2.32 (0.14–37.22)
*IL-18* ^#^	C	G	C	(52.5)	(53.3)	0.81	0.96 (0.68–1.36)
	A	C	G	(25.7)	(27.2)	0.67	0.92 (0.62–1.36)
	A	G	C	(13.3)	(8.7)	0.08	1.66 (0.96–2.89)
	C	G	G	(4.1)	(5.7)	0.45	0.72 (0.30–1.17)
	A	C	C	(1.2)	(2.1)	0.46	0.57 (0.12–2.71)
	A	G	G	(2.6)	(1.5)	0.40	1.66 (0.52–5.30)
	C	C	G	(0.0)	(1.1)	1.00	0.00 (0.00–0.00)
	C	C	C	(0.6)	(0.4)	0.91	1.15 (0.10–12.75)

CI, confidence interval; IL, interleukin; *N*, number of individuals; OR, odds ratio. ^‡^ Polymorphisms in haplotypes: *IL-1*: rs1800587/rs1143634/rs2234663; *IL-10*: rs1800896/rs1800871/rs1800872; *IL-18*: rs1946518/rs187238/rs4988359. Haplotypes are sorted according to the decreasing haplotype frequency in controls. ^#^ Two results missing in the group of controls (total *N* = 208) due to poor quality of isolated DNA.

**Table 3 ijms-21-04728-t003:** The presence of bacteria in subgingival sulci/pockets in the subgroups of patients with the aggressive form of periodontitis (AgP, cases) and healthy controls.

Bacteria	Presence	Patients with AgP *N* = 25 (%)	Controls *N* = 51 (%)	*p*	OR 95% CI	Non-Smokers with AgP	Non-Smokers Controls	*p*	OR 95%CI
*A. actinomycetemcomitans*	negative	16 (64.0)	37 (72.5)			10 (71.4)	33 (73.3)		
	positive	9 (36.0)	14 (27.5)	0.31	1.49 (0.53–4.13)	4 (28.6)	12 (26.7)	0.57	1.10 (0.29–4.18)
*T. forsythia*	negative	1 (4.0)	23 (45.1)			1 (7.1)	21 (46.7)		
	positive	24 (96.0)	28 (54.9)	0.000137 *	19.71 (2.48–157.03)	13 (92.9)	24 (53.3)	0.0064	11.38 (1.37–94.45)
*P. gingivalis*	negative	4 (16.0)	42 (82.4)			1 (7.1)	37 (82.2)		
	positive	21 (84.0)	9 (17.6)	<0.000001 *	24.50 (6.75–88.92)	13 (92.9)	8 (17.8)	0.000001	60.13 (6.85–528.08)
*T. denticola*	negative	3 (12.0)	28 (54.9)			1 (7.1)	25 (55.6)		
	positive	22 (88.0)	23 (45.1)	0.000262 *	8.93 (2.37–33.63)	13 (92.9)	20 (44.4)	0.0012	16.25 (1.96–135.01)
*P. micra*	negative	3 (12.0)	22 (43.1)			3 (21.4)	20 (44.4)		
	positive	22 (88.0)	29 (56.9)	0.005410 *	5.56 (1.48–20.98)	11 (78.6)	25 (55.6)	0.11	2.93 (0.72–11.96)
*P. intermedia*	negative	9 (36.0)	28 (54.9)			4 (28.6)	26 (57.8)		
	positive	16 (64.0)	23 (45.1)	0.10	2.16 (0.81–5.80)	10 (71.4)	19 (42.2)	0.054	3.42 (0.93–12.57)
*F. nucleatum*	negative	0 (0.0)	3 (5.9)			5 (35.7)	3 (6.67)		
	positive	25 (100.0)	48 (94.1)	0.30	1.56 (0.15–15.81)	9 (64.3)	42 (93.3)	0.014	0.13 (0.03–0.64)

CI, confidence interval; *N*, number of individuals; OR, odds ratio; negative, less than 10^3^ colony-forming units (CFU) of the respective bacteria in the sample of subgingival sulci/pockets; positive, more than 10^3^ CFU of the respective bacteria in the sample of subgingival sulci/pockets. * *P*_corr_ < 0.05.

**Table 4 ijms-21-04728-t004:** Concentrations of interleukin-10 (IL-10) in peripheral blood monocytes (PBMC) stimulated by selected periodontal bacteria.

PBMCs	IL-10 Levels (pg/mL)			*p*
	Median	IQR	Minimum–Maximum	
Unstimulated (negative control)	1.8	0.4–6.2	0–13.9	
With PWM (positive control)	85.7	61.7–302.0	14.4–3447.0	0.00013
Stimulated by bacteria				
*A. actinomycetemcomitans*	30.1	15.9–136.9	4.4–663.1	0.00009
*T. forsythia*	75.9	29.2–173.6	0.6–320.1	0.00035
*P. gingivalis*	0.7	0.2–5.0	0.0–45.7	0.287
*P. intermedia*	5.3	2.3–13.0	0.0–173.6	0.034 (*P*_corr_ > 0.05)

IQR, interquartile range; PWM, pokeweed mitogen; calculated by Willcoxon pair test.

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
