# Peer review of "Interleukin Gene Variability and Periodontal Bacteria in Patients with Generalized Aggressive Form of Periodontitis"

_ijms, 2020, doi:10.3390/ijms21134728_

Round 1
Reviewer 1 Report
[Suggestions]
Minor;
L. 134-135, and Table 2:
"the IL-10 GCC (rs1800896/ rs1800871/ rs1800872) haplotype was associated with AgP in the recessive model (P = 0.02...)"
The referee is unable to find the date of P = 0.02 in the Table 2.
L. 135-137, and Table 2:
"the IL-18 AGC (rs1946518/rs187238/rs4988359) haplotype was present significantly more often in patients with AgP in the dominant model (P = 0.046....).
Again, the referee is unable to find the date of P = 0.046 in the Table 2.
L. 171-172, and Table 3:
"(96.0% vs. 66.7%; very high risk was determined in 68.0% AgP vs. 13.7% in controls, 171 respectively, P = 0.0032)."
The referee is unable to find these data in the Table 3.
L. 252-274:
Regarding the IL-18, the new index such as "3.2. Interleukin-18", independent of "3.1. Interleukin-1 Family", could be created in the manuscript.
Typographical errors:
L. 107: "bweree" and "bewas" should read "be".
L. 108: "anthe" should read "the".
L. 387, and 394: "Figures A and C" should be "Figures 4A and 4C".
L. 397: "Figures B and D" should be "Figures 4B and 4D".
L. 400: "Figure C" should be "Figure 4C".
L. 591: "T. forsythia, P. gingivalis, T. denticola .... P. micra" should be in italics.
Author Response
To: Reviewer 1
Comment 1: Minor suggestions:
L. 134-135, and Table 2:
"the IL-10 GCC (rs1800896/ rs1800871/ rs1800872) haplotype was associated with AgP in the recessive model (P = 0.02...)"
The referee is unable to find the date of P = 0.02 in the Table 2.
- 135-137, and Table 2:
"the IL-18 AGC (rs1946518/rs187238/rs4988359) haplotype was present significantly more often in patients with AgP in the dominant model (P = 0.046....).
Again, the referee is unable to find the date of P = 0.046 in the Table 2.
Answer 1: Table 2 shows only results of the multilocus haplotype analysis calculated according to the multiple model (which is best suited for the multifactorial genetics of periodontitis). The results on lines 134-137 are results calculated using the recessive or dominant model for haplotype analysis (but after correction for multiple comparisons were insignificant) were mentioned in the text only.
We believe that adding the mostly insignificant results from other statistical models into Table 2 would not help the clarity of the table and would make it overcomplicated and difficult to read. However, we appreciate that this was not stated clearly enough in the manuscript and, therefore, we amended the manuscript as follows:
“Frequencies of all haplotypes between cases and controls (as calculated by a multiple model) were similar, see Table 2. Nevertheless, in the recessive model, the IL-10 GCC (rs1800896/ rs1800871/ rs1800872) haplotype was associated with AgP (P = 0.02, OR = 2.07, 95% CI = 1.13–3.78, Pcorr > 0.05) and in the dominant model, the IL-18 AGC (rs1946518/rs187238/rs4988359) haplotype was present significantly more often in patients with AgP (P = 0.046, OR = 1.84, 95% CI = 1.02–3.34, Pcorr > 0.05).”
We also changed the description of Table 2 as follows:
“Table 2. Haplotype frequencies of selected interleukin polymorphisms in patients with the aggressive form of periodontitis (AgP) and healthy controls and the evaluation of the differences between AgP and control groups (P and OR) calculated using the multiple model.”
Comment 2: L. 171-172, and Table 3:
"(96.0% vs. 66.7%; very high risk was determined in 68.0% AgP vs. 13.7% in controls, 171 respectively, P = 0.0032)."
The referee is unable to find these data in the Table 3.
Answer 2: The Reviewer is right, this information is not in the Table 3. The reason is that Table 3 only contains “hard data” while the sentence in question relays a hypotetical result implicated by the micriobial findings. The risk score of periodontal disease for each patient is calculated by Protean Ltd., Ceske Budejovice, Czech Republic, using their original proprietary algorithm based on the presence and relative quantity of the periodontal bacteria in samples obtained from periodontal pockets/sulci. The company, however, did not disclose their methods to us. This information was added into the Material and Methods part.
“The hypothetical risk score of periodontal disease (low/increased/high/very high risk) was calculated by Protean Ltd., Ceske Budejovice, Czech Republic, using their original proprietary algorithm based on the presence and relative quantity of the periodontal bacteria in samples obtained from periodontal pockets/sulci.”
We have also clarified the paragraph to read as follows:
“The risk rate for periodontal diseases calculated from the microbial assessment of the presence/absence of all seven investigated bacteria was higher in the patients with AgP than in controls (96.0% vs. 66.7% of patients with increased, high or very high risk; very high risk was found in 68.0% AgP vs. 13.7% in controls, respectively, P = 0.0032).”
If, however, the Reviewer or the Editor are not happy with publishing a result that was obtained using a proprietary algorithm that is not fully disclosed, we are willing to completely remove these two paragraphs from the manuscript.
Comment 3: L. 252-274:
Regarding the IL-18, the new index such as "3.2. Interleukin-18", independent of "3.1. Interleukin-1 Family", could be created in the manuscript.
Answer 3: IL-1 family ligands include 7 molecules with agonist activity (IL-1α, IL-1β, IL-18, IL-33, IL-36α, β, and γ), three receptor antagonists (IL-1Ra, IL-36Ra and IL-38), and an anti-inflammatory cytokine (IL-37). For this reason, we believe that the results of IL-18 analysis should indeed be detailed in the manuscript together with those of IL-1α, IL-1β and IL-1RN.
However, if the Reviewer insists, we could possibly create subchapters for the 3.1.Chapter but then, to be consistent, we would have to create a subchapter for each of those above mentioned IL-1 family members (i.e., 3.1.1. IL-1α; 3.1.2 IL-1β, etc.) and it would in our opinion disrupt the flow of the manuscript.
Comment 4: Typographical errors:
L. 107: "bweree" and "bewas" should read "be".
L. 108: "anthe" should read "the".
L. 387, and 394: "Figures A and C" should be "Figures 4A and 4C".
L. 397: "Figures B and D" should be "Figures 4B and 4D".
L. 400: "Figure C" should be "Figure 4C".
L. 591: "T. forsythia, P. gingivalis, T. denticola .... P. micra" should be in italics.
Answer 4: Thank you very much for your comments, errors were corrected.

Reviewer 2 Report
The present ms. entitled "Gene variability in interleukins and periodontal bacteria in Czech patients with generalized aggressive form of periodontitis" by Drs. Holla and colleagues is an important study related to the pathogenesis and etiology of aggressive periodontitis. The authors demonstrate that altered IL-10 levels and specific bacteria are responsible for lymphocyte balance and resulting aggressive periodontitis phenotypes. This is a well written and well documented study that will greatly contribute to the periodontal literature and especially to this Special Issue. Figures are excellent and statistics are well executed.
This reviewer has no concerns. An excellent manuscript!
Author Response
To: Reviewer 2
Comment 1: The present ms. entitled "Gene variability in interleukins and periodontal bacteria in Czech patients with generalized aggressive form of periodontitis" by Drs. Holla and colleagues is an important study related to the pathogenesis and etiology of aggressive periodontitis. The authors demonstrate that altered IL-10 levels and specific bacteria are responsible for lymphocyte balance and resulting aggressive periodontitis phenotypes. This is a well written and well documented study that will greatly contribute to the periodontal literature and especially to this Special Issue. Figures are excellent and statistics are well executed.
This reviewer has no concerns. An excellent manuscript!
Answer 1: Thank you very much for your positive evaluation of our work.
